# Identifying Home System of Practices for Energy Use with K-Means Clustering Techniques

**Troy Malatesta \* and Jessica K. Breadsell**

Sustainability Policy Institute, School of Design and Built Environment, Curtin University, Building 209, Level 1, Kent St., Bentley, WA 6102, Australia; jessica.breadsell@curtin.edu.au
\* Correspondence: troy.malatesta@postgrad.curtin.edu.au

**Abstract:** Human behaviour is a major driver and determinant of household energy consumption, with routines and practices shaping daily energy profiles. These routines and practices are made up of individual lifestyles and other contextual factors that vary from home to home. Social and psychological theories aim to explain and describe how people consume resources in the home, which has resulted in the development of the home system of practice. This evaluates how occupants live and follow multiple routines which result in varying energy consumption practices. This paper develops a methodology to identify and support the concept of the home system of practice using a data analytical approach and link it to residential energy and distribution network management. This paper utilises k-means cluster analysis to identify these different home systems of practices and routines in energy use by using real-time energy consumption data from July 2019 to March 2021 from a living laboratory in Australia. The results of the analysis show the different daily energy profiles for each of the 39 households, with some homes observing large fluctuations and changes in the way they consume energy during the day. Specific homes were discussed as case studies in this paper focusing on linking the occupants' contextual factors to their energy profiles. This variation is discussed in terms of the routines of the occupants and associated lifestyles that explain why some energy peaks occurred at different parts of the day and differed during the COVID-19 lockdown period in Australia. The paper conducts a comparison between these case studies to show how people's lifestyles impact household energy consumption (and variation). These case studies investigated the heating and cooling practices of the occupants to demonstrate how they impact overall consumption. This variation is discussed in relation to energy management and prediction of when homes will consume energy to assist in net-zero energy developments and grid stabilisation operations.

**Keywords:** home system of practice; net-zero energy home; automation; energy management; social practice theory; behaviour; machine learning

## 1. Introduction

The residential sector is undertaking an energy transition moving away from carbon intensive technologies in response to the rise of carbon emissions and climate change [1]. This change in technology has resulted in challenges and barriers that need to be overcome to achieve a successful energy shift. Renewable technology offers a clean method of producing energy; however, there are many issues with energy reliability and alignment with renewable energy generation which can result in supply not aligning with demand [2]. Additionally, the electrification of heating, cooling, cooking, and transportation is forecasting an increase in energy demand from the residential sector [3]. Thus, the way households consume energy must be studied to understand how policies can impact this consumption and develop systems to manage energy supply and demand [4].

An approach that can assist in developing policies and encouraging the optimisation of renewable technology is to understand from a social perspective how people in homes consume energy. This entails understanding the behaviour, practices, habits, and routines

of occupants [5]. These routines are often repetitive and can correlate strongly to the energy consumption profile of the home. This allows researchers to understand the temporal characteristic of energy usage [6]. This temporal characteristic is an important factor to consider when developing and incorporating renewable, often intermittent energy technologies into the energy mix. Thus, understanding when energy will be consumed will allow for better energy management, achieving higher grid reliability and security.

The routinised nature of individual lifestyles and energy usage allows data analytics to play a role in identifying patterns [7]. Households can follow different routines depending on seasonal and institutional characteristics, including following different heating and cooling routines in the summer and winter [8]. The ability to understand how these routines vary throughout the year allows researchers to predict energy consumption. The uptake of home sensors and monitoring has increased the amount of data on energy behaviours and consumption routines throughout the day. The utilisation of machine learning algorithms and cluster analysis can identify these routines and assist in energy management and understanding the metabolic nature of energy consumption within the home [9].

This paper utilises a k-means clustering approach to identify trends and patterns in energy data collected from an Australian residential living laboratory. This analysis is aimed at supporting social theories regarding human behaviour and practices within the home and the concept of the home system of practice (HSOP). The novelty of this research relates to supporting social theories with a robust data analytical methodology. The k-means approach aims to complement the principles of social theories relating to occupant behaviour and household energy consumption by identifying this behaviour in energy data. This will assist in assessing different household routines, including the temporal aspect. In conclusion, this paper discusses how an HSOP can be identified through data analysis and how different HSOPs can be evaluated to assess how routinised a household is.

## 2. Literature Review

### 2.1. Home System of Practice

Insight from studying the social aspect of energy consumption is important during the development of energy policies. The focus has been shifted to the study of the societal factors of energy consumption and understanding how energy use is constituted socially and materially [10–18]. This is accompanied by understanding the attitudes, behaviours, and choices of individuals that relate to the context of energy consumption [19,20]. These behaviours correlate to the practices performed by individuals in order to achieve certain social outcomes. Each individual or family has unique routines and habits that result in variation in the practices that are performed. Studying how energy is consumed in the home and considering the home as a physical system that involves resource flows in and out of the system can explain this variation in use [21,22].

Psychological and social theories have been utilised to explain resource consumption within the home for decades. This allows researchers to understand how individuals use energy in their daily lives and assess their everyday routines. These theories outline how humans live in a routinised way and perform similar activities at similar times [23]. This is summarised within social practice theory, which offers a basis for assessing the performance of household practices. The routines and repetitive nature are a result of people performing half-conscious activities that result in resource consumption [23]. They are composed of several elements: the technology necessary to perform the task, the skills and understanding to perform it successfully, and the motivation or meaning driving the desire to act in that way. This theory is developed by outlining that individuals perform practices to achieve a social outcome that results in resource consumption [24]. For example, the practice of showering can be performed by an individual to achieve a sense of cleanliness, which results in water being consumed [25].

This approach allows researchers to analyse and understand how to achieve a reduction in resource consumption through assessing people's interactions with infrastructure and social systems [26]. These interactions and interrelationships can be evaluated to

provide insight into the performance of certain practices that relate to resource consumption [27–29]. Many papers have discussed this practice-based approach and how these practices are related to the routines of an individual [30,31].

A household can consist of a single individual or multiple individuals that follow different routines throughout the day. These routines are made of up practices performed by an individual that result in resource consumption. The combination of multiple routines within the household creates a home equilibrium that allows everyone to perform their desired practices without inhibiting anyone else. This home equilibrium refers to the HSOP that considers how people's lifestyle and routines interlink and locks in practices to a temporal field [21,32–36].

Once the HSOP is developed and routines are created, the expectation is to see consistent and regular energy consumption profiles. The complexity of the HSOP and the interaction between occupants can lock individuals into a certain lifestyle where practices need to be performed at certain times of the day [37,38]. The inherent flexibility of this HSOP is dependent on the lifestyles, attitudes, and beliefs of the occupants [39]. For example, a highly flexible household can be achieved by having a single occupant working part-time, allowing the occupant to perform household activities at different times of the day throughout the week. However, a locked-in household is observed with a typical family home (two adults and one child) where each adult is working full time (9:00 a.m. to 5:00 p.m.), and the child attends school during the week. This results in practices occurring at the same time every day to fit around the working and schooling lifestyles of the occupants [32]. The comparison between the locked-in and the highly flexible homes shows how the context of the individuals impacts how each home consumes energy. The timing of energy-consuming practices in locked-in households may be shifted by the use of timers on appliances (such as hot water systems or dishwashers), but only if the required technology, skill, and desire are present.

### 2.2. Energy Demand Profiles

The HSOP's relation to the energy profiles of a household can be studied to understand the temporal characteristics of peak daily loads. These load curves demonstrate the daily, monthly, and seasonal variations in how, where, and when energy is consumed [40]. Typical annual load curves exhibit seasonal peaks and troughs, while daily and weekly curves observe a saw-toothed shape outlining the fluctuations in energy demand throughout the day, peaking during the morning and late afternoon [41]. Power systems and networks are managed to ensure sufficient capacity and supply to meet this variation in demand to achieve network reliability [42,43]. The decarbonisation of the power network has presented several challenges including the maintenance of the power system and network reliability to ensure energy demand is met. Thus, the management of power and the associated peak demand is crucial in ensuring affordable and reliable energy provisions are generated by low-carbon technologies.

Renewable technologies offer a potential strategy for decarbonising the residential distribution network and achieving emission reductions. Output management of these technologies involves crude methods (e.g., turning off wind turbines) or storage requirements (e.g., batteries) which result in significant financial impacts [40]. In addition, the alignment between energy generation from solar and wind sources and the typical 24 h demand profile is limited, with the majority of peak demand occurring outside of this generation period. This misalignment is understood through the concept of the HSOP as a misalignment between the timing of the energy consumption practices of the home and renewable energy generation profiles.

The HSOP is a well-established equilibrium that works around and supports all the household's scheduled events such as work and school [21]. These events can be categorised as social and institutional rhythms that control the lifestyle of individuals and relate to when individuals will be occupying the home [31]. The occupancy of the home relates to the timing of energy consumption, as when no one is home, there will be limited energy

demand from the home (unless automation systems are utilised). Institutional rhythms such as working hours dominate a person's activities during the week and will create a routine, often from Monday to Friday, resulting in a typical household energy profile [31]. A typical home energy profile observes a morning peak when people are getting ready for work and a late afternoon peak representing when people return from work [40]. This profile is common between households, as the majority of people share these practices and have similar working lifestyles; hence, the distribution network is exposed to high peaks during the morning and afternoon. There are also expected changes during public holiday periods, school holidays, and extreme weather events. It remains to be seen whether the changes to working patterns seen during the COVID-19 pandemic will be long-lasting or whether usual practices will return once the pandemic lockdowns and work-from-home practices subside [44].

### 2.3. Energy Management

Understanding the timing of energy consumption is the first step to achieving efficient household energy management. Knowledge of the different routines and practices performed by the occupants will allow systems to predict the magnitude of energy required to allow energy supply to be manipulated and dynamic. An important aspect of residential living is the effective use of energy [45], and thus the ability to predict energy consumption will help to achieve this. Managing electricity generation and consumption provides avenues for more efficient methods to be used to achieve energy savings, resulting in an environmental benefit through reduced emissions [45]. Previous studies revealed how effective energy management methods can be in commercial and residential settings [46]. These methods involved the utilisation of renewable energy and energy-saving techniques to allow hotels gain green building energy certifications. Other methods, including machine learning (including clustering approaches) and identifying patterns in energy consumption, have been used to assist energy management systems in understanding the variation in household routines [47–50]. Additionally, other research has highlighted the potential of education and changing individuals' attitudes and mindsets to achieve energy savings [51,52]. However, achieving long-term behavioural changes is difficult, with many individuals reverting to their previous behaviours [32]. This is the challenge of using social practice theory and achieving long-term energy savings in the residential sector.

Other energy management and reduction methods involve changing and utilising new technology, with a study demonstrating significant energy savings by upgrading air-conditioning systems to newer, more efficient models [53]. Instead of aiming to impart behavioural change, this approach relates to social practice theory and changing an element of the practice (in this case the technology used) to automatically achieve energy savings without the need for upskilling or behavioural change. These methods show the movement towards energy management within the household and how this study aligns with the trends in research.

### 2.4. Research Gap

The current gap in research is the lack of data analytical approaches to support psychological and social theories that try to explain how and why energy is consumed within the household. Past research used manual and/or simplistic data analytical methods to assess the HSOP and identify household behaviours [21,32,33]. The use of clustering algorithms is common in research [54–56] to identify consumption patterns; however, such algorithms and results are not discussed or linked to these theories. This paper aims to fill this gap by developing a data analytical method to support the HSOP and these theories.

## 3. Methods
### 3.1. The Living Laboratory

The Fairwater Living Laboratory project consists of a sustainable housing precinct in Australia that incorporates ground-source heat pumps to replace typical energy-intensive

heating and cooling systems. This project is Australia's largest precinct-scale installation of ground source heat pumps in a residential setting [57]. The precinct consists of 850 homes, with a subset of 39 homes being selected to be monitored as part of a living laboratory study between 2019 and 2021. The precinct is located in the suburb of Blacktown, Sydney, New South Wales, Australia, and the development was awarded the top 6-star Green Star Communities rating under the Green Building Council of Australia's accreditation scheme.

The living laboratory utilised a multi-method approach to study the social and technical data of the occupants and to understand how energy is consumed within the home. The homes within this subset ranged from two- to five-bedroom houses, with all homes in this study being owner-occupied and with characteristics displayed in Table 1. All homes had their electrical energy use at a circuit level monitored with environmental monitoring that included indoor and outdoor temperature, humidity, and other conditions.

**Table 1.** Summary of the house designs that were a part of this study.

| Type of Home | Number of Homes | Number of Residents | | | | Internal Area (m²) | | | | NatHERS Star Rating | | | |
|---|---|---|---|---|---|---|---|---|---|---|---|---|---|
| | | Min | Max | Median | Std. Dev. | Min | Max | Median | Std. Dev. | Min | Max | Median | Std. Dev. |
| 2-Bedroom | 7 | 1 | 5 | 3 | 1.2 | 83 | 131 | 112 | 13.3 | 4.0 | 7.4 | 5.5 | 1.1 |
| 3-Bedroom | 12 | 1 | 3 | 3 | 0.7 | 126 | 191 | 156 | 11.7 | 4.5 | 6.0 | 5 | 0.6 |
| 4-Bedroom | 19 | 2 | 5 | 4 | 0.9 | 168 | 291 | 215 | 15.1 | 3.8 | 5.5 | 5 | 0.8 |
| 5-Bedroom | 2 | 3 | 7 | 5 | 2 | 270 | 285 | 277 | 7.5 | 4.0 | 4.5 | 4.3 | 0.25 |

## 3.2. Data Collection and Analysis

For this study, the electrical energy use of the homes was the focus of the analysis to identify the patterns of energy consumption for each individual home. The energy data were collected at 30 min intervals, and the circuits that were monitored were the mains electricity, air conditioning, lights, power, oven, solar inverter (if applicable), water tank, garage, and loft (if applicable). The electrical data were collected between 1 July 2019 and 31 March 2021 with minimal gaps in the dataset, resulting in minimal data cleansing being required. The sum of these circuits was used to investigate the patterns of energy consumption and to gain insight into the everyday practices of the homes. This period did include the COVID-19 pandemic lockdowns, which may have changed some household energy and work practices. The majority of restrictions that occurred in Sydney during this timeframe were between 23 March 2021 and 2 June 2021. The impact of these restrictions is considered in the analysis of the results.

The purpose of this research is to assess the consumption behaviour patterns of the occupants and identify households that exhibit variation in the way they consume energy. This paper uses a clustering approach to develop a household's typical consumption curve(s) throughout the study period. Clustering is the process of dividing a dataset into several subsets, with each subset being different to the other subsets, and objects in each subset will have similarities [58]. The goal of this clustering approach is to maximise the similarities and distinguish the differences within the dataset.

The energy consumption data were analysed through a k-means clustering algorithm to recognise patterns in the dataset. This technique is often use for energy datasets and utilised in the electricity market to identify variation in energy profiles and to forecast future demand [59–65]. The k-means algorithm is an unsupervised approach to partitioning and grouping data into clusters that aims to minimise the distance between all points and their cluster centre [66,67]. Unsupervised learning involves analysing and clustering unlabelled clusters without the need for user intervention. Additionally, it is crucial to minimise this intra- and inter-cluster distance to ensure points are classified accurately with other similar points and that all clusters have homogenous subsets.

The k-means clustering steps are as follows:

1. Randomly select 'k' as the number of initial cluster centres
2. Calculate the distance from each data object to each cluster centre and assign the objects to the clusters with the closest distance
3. Assign all data objects and recalculate the centres of all the clusters
4. Iterate Steps 2 and 3 until data objects are being assigned to the same cluster without any changes.
5. Output the clusters

A detailed explanation and step through of the k-means algorithm and iterative steps can be found in [58].

This approach is impacted by the selection of the number of clusters for the algorithm to split the data into. This can be done in numerous ways. This research is creating a relationship between the number of distinct clusters and the HSOP; hence, this pre-defined number must be selected appropriately. This paper used the NbClust package in R to determine the best number of clusters [68]. This analysis was complemented by visual inspections of the final clusters to confirm the package has identified unique clusters. This visual inspection provided a sanity check for the results before relating them to the HSOP and occupant behaviour.

This machine learning approach was used to identify clusters for individual households and relate these clusters to social practice theory and the HSOP, as shown in Figure 1. Furthermore, the study used the basis that these clusters and the number of clusters identified related to the nature of the routines and how repetitive each household is in their practices and behaviours. A higher number of clusters would translate to the household being less routinised and more irregular in their energy consumption, resulting a dynamic HSOP. The energy consumption of such households will be difficult to predict, and furthermore, it will be difficult to develop energy management and automation systems to assist and complement their energy consumption. Each cluster relates to a specific HSOP that is followed by the household, and this cluster relates to a specific temporal context. This demonstrates how practices and energy consumption vary depending on the time of year, seasonal differences, and other contextual factors (such as a COVID-19 lockdown).

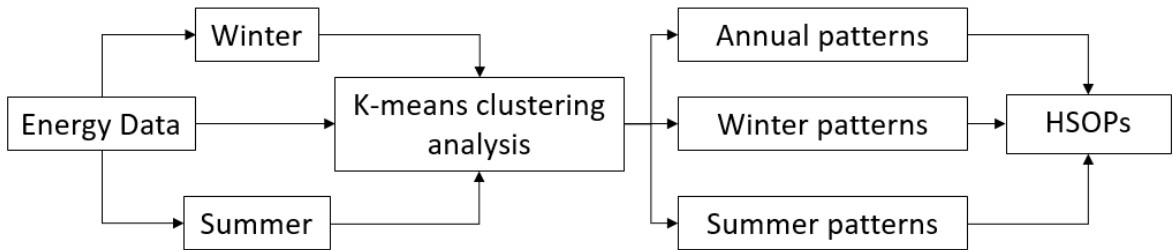

**Figure 1.** Flow diagram of the methodology followed to identify patterns in energy consumption.

## 4. Results and Discussion

This study hypothesised that a machine learning clustering approach can be taken to support social practice theory and the concept of the HSOP. Pattern identification in energy data can demonstrate the routinised practices of individuals and provide insight into how energy is consumed. Furthermore, it describes how some households vary significantly in their energy consumption compared to other, more routinised households. This can be linked to home energy management and understanding when peak demand periods will occur.

### 4.1. General Energy Consumption

The analysis began with evaluating house performance and energy consumption across the all the living laboratory study houses to provide a baseline for individual home comparison. The daily average consumption profile was generated from the energy dataset

by simply averaging all the homes' daily consumption profiles. This is shown in Figure 2. And demonstrates the general timing of consumption for the households. On average, the homes follow a dual-peak consumption profile with a peak in the morning and a peak in the late afternoon. This reveals the dominant routines of the living laboratory and identifies possible institutional rhythms that the occupants follow. These institutional rhythms include going to work and/or school during the day. Hence, there is a decrease in consumption during the day. This profile follows typical energy consumption patterns seen in urban areas around the developed world and offers an initial insight into how routines and practices can shape a household's energy consumption. Further analysis on individual homes will demonstrate more specific routines and identify the practices and behaviours that shape a household's daily energy consumption that are not captured by this generalised analysis.

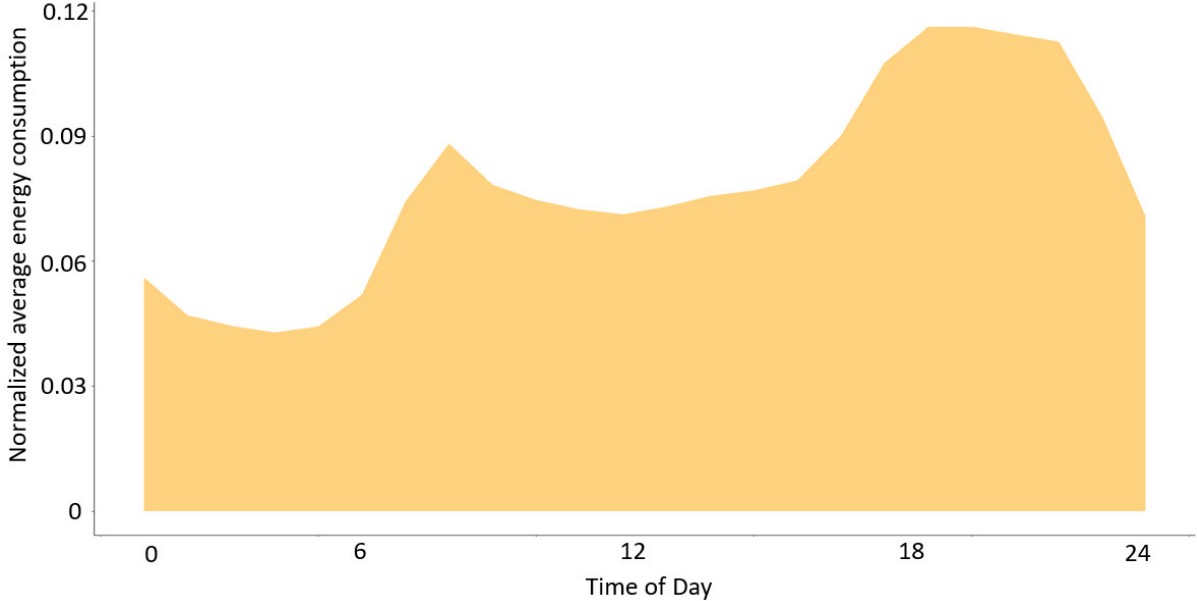

**Figure 2.** Average daily energy profile from all homes in the study.

The 39-home energy dataset was analysed using the k-means algorithm to identify typical daily energy consumption profiles that are followed by all homes (Figure 3). This demonstrates how practices can be socially shared between homes, resulting in an aggregated peak demand period for the precinct. Specifically, the winter and summer energy profiles were extracted from this dataset to outline how the living laboratory performs differently in these seasons. The results from this analysis will allow for a comparable view when studying the homes on an individual level.

Overall, the k-means analysis identified eight different clusters for the 39 homes, which highlight the typical daily energy consumption profiles of the precinct as a whole. Each cluster offers insight into a specific shared routine of the precinct that is followed throughout the year. This determines the variability of energy consumption as a whole and provides insight into the practices, behaviours, and habits of the precinct occupants. Analysis of these clusters can show how the shape and temporal characteristic of the peaks of the profile change depending on the day. The timing of the peaks is variable between clusters, providing insight into the dynamic nature of energy consumption across all the different households [69,70]. This provides insight for a large-scale distribution network to monitor and predict the energy demand from these homes and into how energy operators can manage their supply to respond to this variability in demand. Further discussion on this topic will be included later in this paper.

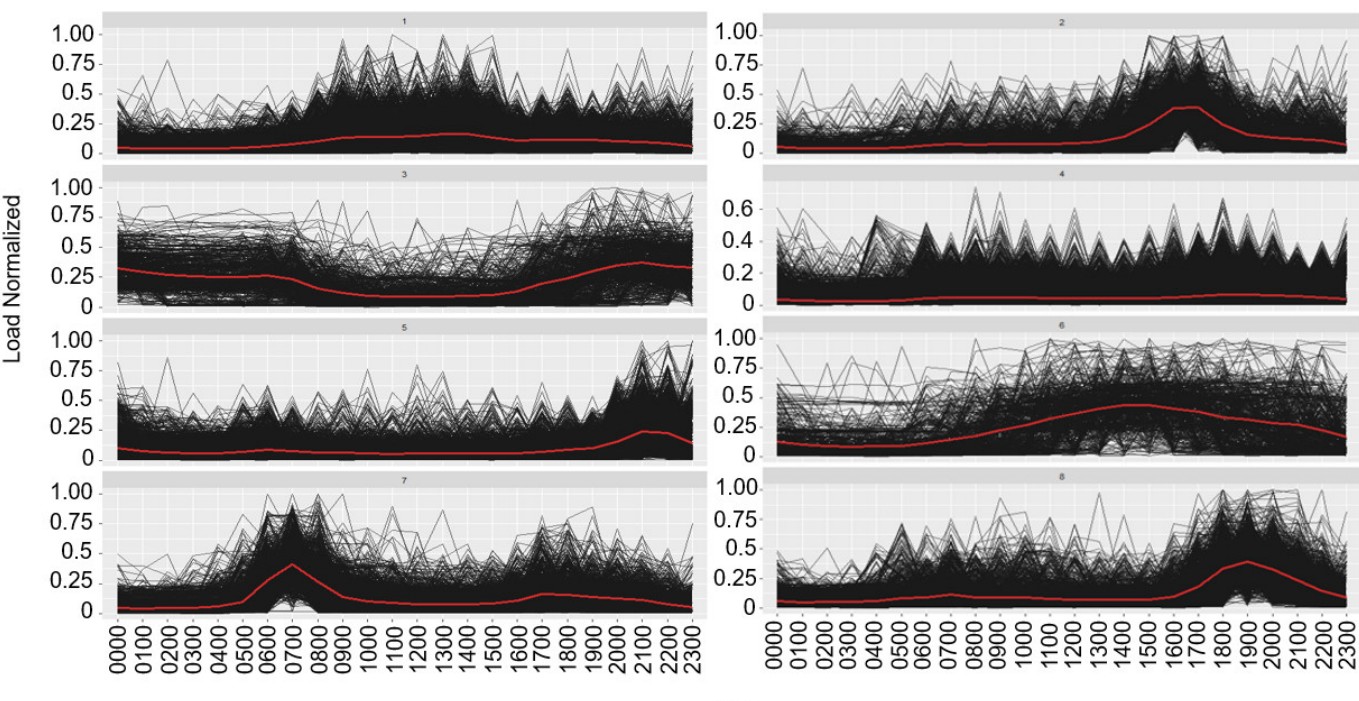

**Figure 3.** K-means analysis results for all the homes, incorporating energy data from all homes and identifying energy patterns for the whole study. The black lines represent the data objects that were assigned to that cluster and the red lines represent the energy profile for that cluster.

### 4.2. Pattern Identification

The dataset was separated to conduct the k-means analysis on the homes individually to study the routines and the HSOP throughout the study period. The number of optimal clusters provides insight into the repetitive nature of the occupants and determines the variability of the HSOP. A higher number of clusters would show the household is very dynamic in their energy consumption and does not follow strict routines, resulting in several HSOPs.

The results showed a large range in the number of optimal clusters identified by the algorithm. This reinforces and the variability of the HSOP between households and demonstrates the complex nature of occupants' routines and behaviours [31,32,71]. The range of clusters that was identified by the algorithm was between two and nine. There were three households that demonstrated nine different typical daily energy profiles, with one of them shown in Figure 4. As an example.

A visual inspection of these clusters shows the temporal variability of energy consumption for the household. The red line represents the average daily consumption profile for that cluster. The comparison of the peaks across all the clusters outlines the variability of consumption for the household in question. The time of day in which the peak occurs is different between all nine clusters. Furthermore, the shape and magnitude of the peaks offer an insight into the flexibility of the occupants' routines.

This result can be linked back to the HSOP and social theories to provide insight into the practices of the occupants [21]. The data show this household follows nine different HSOPs, with each system having a unique daily energy consumption profile. Some clusters have peaks during the day when occupants would typically be at work; however, for this cluster, the data show the occupants must have changed their typically routines. This provides insight into the variable lifestyle of occupants with different work commitments (e.g., working from home) or family structures (e.g., having children) [6,72]. These changes would explain the increase in energy consumption during the day as a response to a change in the occupancy pattern.

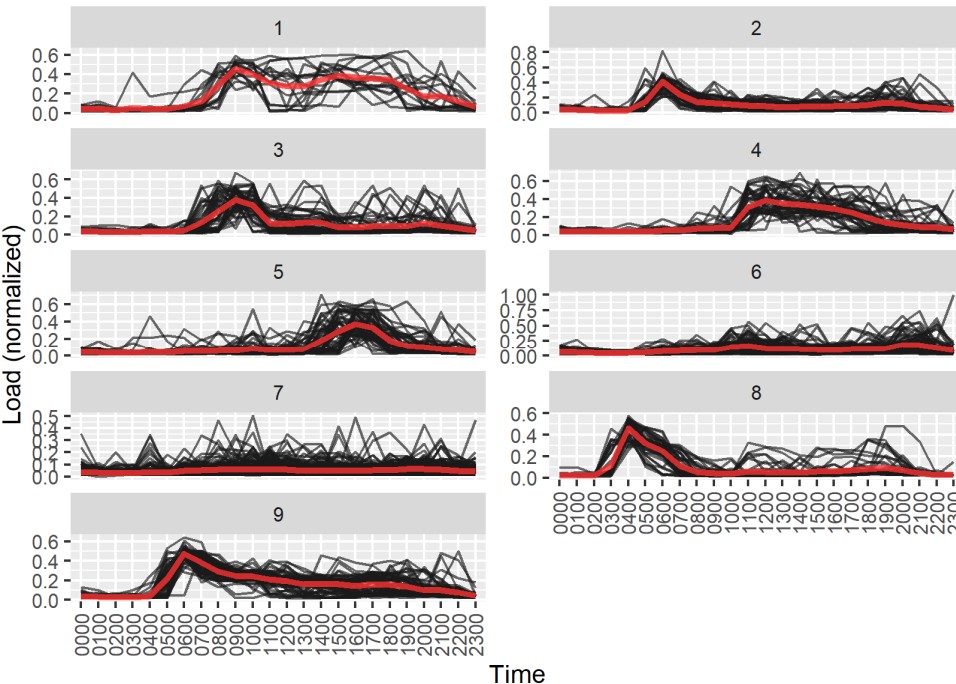

**Figure 4.** House A K-mean results using all year energy data. The black lines represent the data objects that were assigned to that cluster and the red lines represent the energy profile for that cluster.

The characteristics of the households were determined through a survey that was distributed to all households in the living laboratory. The household shown in Figure 4 self-reported that there were three occupants, two of whom were between 40–59 years of age (one male and one female), and the third person was between 18–24 years of age (female). Additionally, the male adult had a full-time job that required him to work away from home. The female adult reported she worked from home, and the young female was a full-time student. The clusters that consist of peaks during the day can be explained by the female adult and young female. Working from home and studying full-time can result in household occupancy during the day that is not observed with typical working lifestyles [73,74]. This can result in higher energy consumption during the day compared to a household with all occupants working away from home. Furthermore, studying full-time can result in variability, as class schedules are not the same every day. This will result in the young female's routines varying significantly, and her occupancy of the home will be irregular, resulting in variable energy consumption.

These characteristics and lifestyles will impact an individual's system of practice and routines, resulting in the HSOP changing. Any changes to an occupant's behaviour will result in a change in the HSOP [32]. Another example of a household that does not follow a routinised lifestyle and a standardised HSOP is shown in Figure 5. This household follows nine different daily energy profiles revealing nine different HSOPs and daily routines. A visual inspection of Figure 5 can reinforce this variability through study of the temporal characteristics of the peaks for each cluster. These occur very randomly during the day, indicating continuously changing routines of the occupants during the study period.

The household consists of two occupants: one male between 30–39 years of age and one female with her age unspecified. The male is employed full-time and works away from home, while the female works from home. This is linked to the previous discussions relating the variability of energy consumption and the working lifestyles of the occupants. The different peaks identified by the algorithm show how the couple follows multiple daily routines.

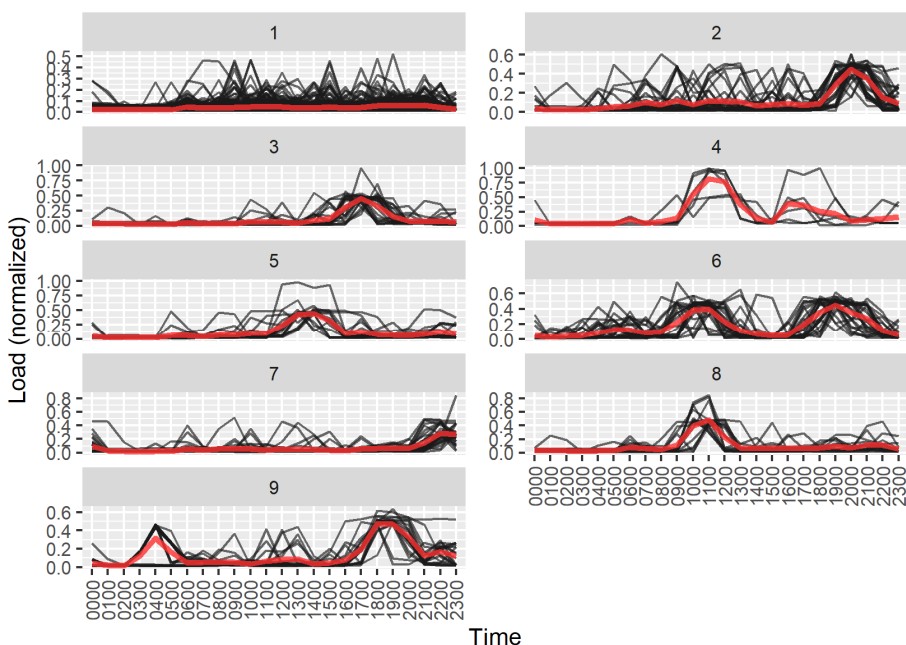

**Figure 5.** House B K-mean results using all year energy data. The black lines represent the data objects that were assigned to that cluster and the red lines represent the energy profile for that cluster.

Many households had fewer identified clusters, indicating a more routinised energy consumption. This reinforces how these households follow a small number of HSOPs and repetitive daily routines. The behaviours and habits of these occupants are not as variable as the previously discussed households. For example, Figure 6 shows a household that follows two distinct daily profiles. The first profile displays a broad peak during the day, representing energy consumption being high between the times 6:00 a.m. and 7:00 p.m. The second profile shows a dual-peak profile with peaks occurring in the morning (between 6:00 a.m. and 8:00 a.m.) and the afternoon (between 5:00 p.m. and 7:00 p.m.). The first cluster with the higher consumption during the day could occur predominantly on the weekend, when home occupation is higher during the day compared to a weekday. This explains why Cluster 2 displays a dip during the day compared to the morning and afternoon peaks, when the occupants are at work during the week. These results align with social practice theory in that the occupants are very routinised, and behaviours are repetitive [6,32,34].

The household consists of five occupants: one male and two females between 30–39 years of age, one male between 18–24 years of age, and one male between 0–4 years of age. The eldest male is employed full-time as a shift worker and works away from home. The two women reported that they work full-time (from home and away from home). All adults are working full-time, which implies their lifestyles would be structured and repetitive, with them following the same daily practices every day. The females sometimes work from home, explaining the first cluster of the household with higher energy consumption during the day, as the home is occupied during these hours.

The number of occupants within the home can impact the flexibility of daily routines, with more restriction in the timing of practices occurring with more occupants. This aligns with the results of House C, which show the five-occupant home only follows two HSOPs. The interaction between occupants creates an equilibrium that allows them to live comfortably together. However, this equilibrium is restricted due to the number of different interactions present within the home.

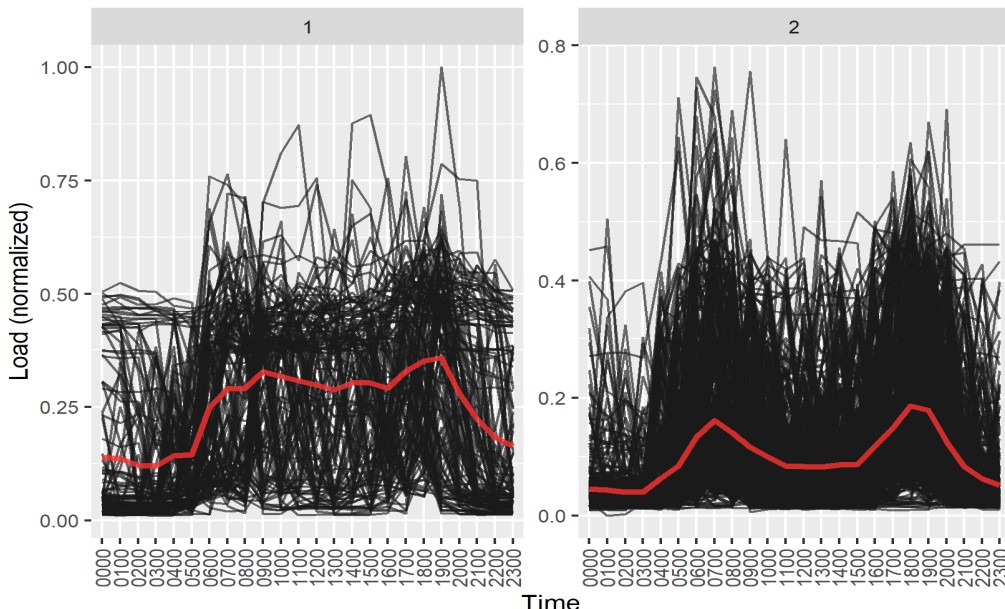

**Figure 6.** House C K-mean results using all year energy data. The black lines represent the data objects that were assigned to that cluster and the red lines represent the energy profile for that cluster.

*4.3. Household Contextual Factors*

The clustering analysis considered the household contextual factors that were collected through conducting surveys. The number of clusters that the households followed ranged between two and nine, and these homes were grouped based off this. Thirty homes out of the forty were assigned between two and five clusters by the algorithm, with the other ten homes being assigned between six and nine clusters, as shown in Table 2. This grouping compares the homes that followed specific routines and the homes that displayed more variation in their energy consumption. The family structure and lifestyle characteristics were extracted from the survey data to see if they explained the large range in clusters between each grouping.

**Table 2.** Summary of the results incorporating household characteristics.

| Clusters | Homes | Average Occupants | Average Occupants Who | |
|---|---|---|---|---|
| | | | Work from Home | Students |
| 2 to 5 | 30 | 3.32 | 1.03 | 0.81 |
| 6 to 9 | 10 | 2.46 | 1.17 | 0.29 |

The first variable considered was the number of occupants in the home and whether this would correlate to the number of clusters. A slight trend was observed: homes with higher clusters often saw a lower average number of occupants. This provides insight into the interaction between individual systems of practices and how they results in variation in the HSOP [31,33]. This aligns with the case study on House C's energy consumption patterns. A preliminary conclusion can be stated that homes with a small number of occupants can have higher variation in energy use. Further investigation is required to support this conclusion. The explanation of this observation can utilise ideas from social practice theory and the idea of flexibility. Large families often have a reduced amount of flexibility in their lifestyles due to all the individual activities and practices that need to be performed [75]. All these activities and practices need to be performed during the day while fitting in with the lifestyles of all the individuals. The interaction among all these individual systems of practices can result in strict routines that can satisfy all individual needs, as shown in House C's cluster analysis [76]. Typical families having

children attending school and adults working during weekdays result in all these practices (such as cleaning, food preparation, entertainment, and general caring for children) being performed in the morning and evening [70]. Additionally, the adults do not have the flexibility to change these routines, and the children's routines are also impacted by their parents' routines. These households do not achieve any flexibility in their routines, which will result in the same HSOP often being followed throughout the year when school and work are scheduled [32]. This would explain why homes with two to five clusters display a higher average number of occupants compared to homes with six to nine clusters.

This preliminary conclusion can be expanded when discussing the correlation between the number of clusters and the number of self-reported students living in a home. Homes with two to five clusters show a higher number of school-aged students within the home compared to homes with six to nine clusters. Previous discussions talked about flexibility and how it impacts the number of HSOPs followed throughout the year [40]. Additionally, the routines of the children are impacted and constrained by the routines of the adults. A higher number of children provides less flexibility for adults to perform different routines and behaviours, and vice versa for the children. This reduced flexibility explains why the analysis calculated a low number of clusters for these homes. The HSOP for these homes is very strict and does not allow much variation in how the occupants can consume energy.

Alternatively, the results show the homes that follow many different daily energy profiles often have a low number of occupants and school-aged students within the home. Building on the discussion, this observation can outline the increased flexibility these households have in comparison, which allows the occupants to perform practices in different temporal spaces and consume energy differently. This is supported by the results from Houses A and B, as the data demonstrate these homes consume energy differently throughout the year and follow many different HSOPs.

The last household factor that was considered was the number of occupants who reported they worked from home during the day. It is important to note that part of these results was collected during the COVID-19 pandemic, which may have impacted the work routines of some occupants. The results show little variation in this number between homes with two to five clusters and homes with six to nine clusters, which implies this factor is not correlated. However, the impact could be hidden and overpowered by the impact of the number of occupants and school-aged students within the home. Further investigation is required to understand the impact working from home has on the HSOP and how energy is consumed within the home.

This discussion provides an overview of how household characteristics impacted the number of clusters and HSOPs identified by the machine learning algorithm. However, there are many more contextual factors that impact the routines and behaviours of individuals that were not discussed in this paper. These factors include the attitudes and beliefs of the occupants and other commitments (such as sporting activities, hobbies) which can impact how they consume resources within the home [5]. A deeper look into the contextual factors of the household and the attitudes of the individuals to further explain the HSOP and results from the clustering analysis should be undertaken in future.

### 4.4. Heating and Cooling Practices

After understanding the impact of household contextual factors on the energy profiles, the next step is investigating the heating and cooling practices of the residents to understand how their daily energy profiles are shaped by this energy-intensive practice [77,78]. The air conditioner use was analysed, and the AC energy data were separated out into winter and summer time periods to evaluate how the residents' daily energy profiles change between these two seasons.

Comparison of the average energy consumption in the winter and summer seasons can outline the different heating and cooling practices of the occupants. The analysis demonstrated the distinct changes in the routines and the way occupants consume energy during these two time periods. Figure 7 outlines the changes to the shape of the daily

consumption profile, with the magnitude and characteristics of each peak being variable. The figure highlights the significant change in consumption between the winter and summer season, with winter heating showing a much higher daily consumption rate. For this study, the winter profile followed a dual-peak profile with significant sharp peaks occurring in the morning and late afternoon. Alternatively, the summer season showed a different shape with no morning peak but a broad medium peak in the afternoon. The magnitude of the afternoon peaks is substantially different, with the summer afternoon peak being approximately 33% lower than the winter afternoon peak. This relates to the way the occupants live differently when exposed to hot and cold seasons and how they consume energy to achieve thermal comfort within the home, which is often linked to the performance of the thermal envelope of the home.

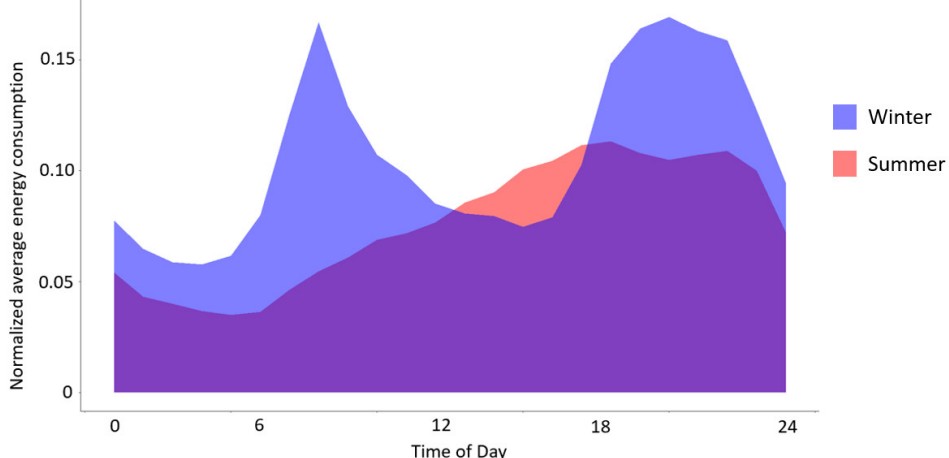

**Figure 7.** Winter and summer average daily profiles for the whole study.

This finding can be linked to social practice theory and the HSOP of the household. The HSOP changes corresponding to external factors such as temporal, seasonal, and environmental conditions [79]. The theory states that occupants perform practices to achieve a desired outcome which results in energy consumption [31]. Thus, the data support previous conclusions that occupants perform different practices in the summer and winter, resulting in varying profiles [80–82]. The use of air-conditioning systems contributes significantly to a household's energy demand; thus, the results focused on the variance in air conditioning use between summer and winter. During the winter, this system is used in the morning and afternoon to achieve thermal comfort, while it is only used in the afternoon during the summer. Additionally, the different magnitudes of the afternoon peaks provide insight into the reliance on the air-conditioning system by the occupants to remain comfortable. This reinforces the variability in people's practices during the summer and winter seasons and reinforces the idea that people live differently in response to varying environmental conditions [83,84].

Clustering analysis was conducted on the summer and winter data for House A to identify the different routines followed during these two seasons, as shown in Figure 8.

House A, which demonstrated nine different profiles and was discussed previously, showed different routines and behaviours between the summer and winter periods. The summer analysis showed the household follows four distinct daily profiles during the summer season. These four clusters are shown in Figure 8, and each peak has its own characteristics regarding time and magnitude. Two clusters observe broad peaks during the day between 8:00 a.m. and 7:00 p.m. which represent when the house is occupied during the day in the summer period. A third cluster shows a sharp peak between 3:00 p.m. and 6:00 p.m. with minimal consumption for the rest of the day. The final cluster displays a random profile where the algorithm was unable to classify the data into groups.

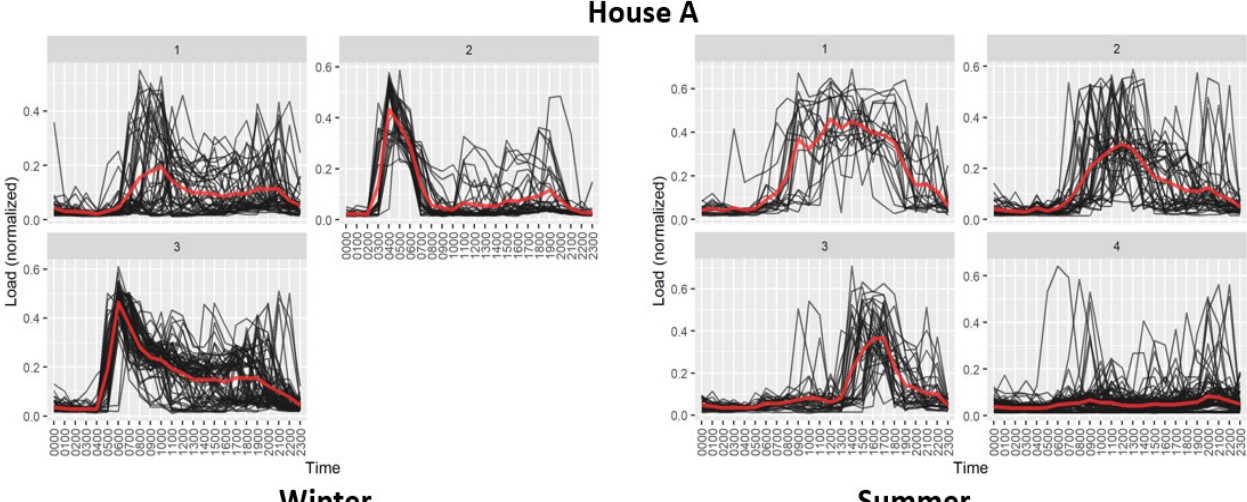

**Figure 8.** House A winter and summer K-mean results. The black lines represent the data objects that were assigned to that cluster and the red lines represent the energy profile for that cluster.

The winter analysis for this household follows three distinct daily profiles, as shown in Figure 8. This displays how the household's energy consumption behaviours vary significantly between the winter and summer periods. Two winter clusters observe sharp peaks in the morning between 4:00 a.m. and 8:00 a.m. and 3:00 a.m. and 7:00 a.m., which is not seen in the summer. This is assumed to be due to the occupants using the heating during the evening and early morning to stay comfortable. The third cluster shows a smaller peak around 8:00 a.m. to 12:00 p.m. The comparison between the two seasons demonstrates how the winter season sees higher consumption during the mornings, while the summer season results in higher consumption in the afternoon. This outlines the variation in how the occupants live and consume energy during hot and cold periods.

House B shows a different picture from House A in terms of summer and winter heating and cooling practices. The algorithm identified nine and seven clusters in the winter and summer periods, respectively, as seen in Figure 9. This outlines the variation in the HSOP of this household and reinforces the results shown previously regarding the clusters identified for the whole year. Visual inspection of the summer clusters shows the algorithm over-defines the clusters, with Clusters 5 and 8 only containing one data point. This can be corrected, however, as the magnitude and timing of the peaks in these clusters are distinct and unique in comparison to the other clusters. This implies that these are outliers that occurred during the year which can be explained by multiple lifestyles and contextual factors such as an anomalous day of activity in the household that did not fit any other past daily routine (e.g., hosting a gathering).

The results confirm the conclusions made previously about the home regarding the irregularity of energy consumption throughout the year. The occupants follow multiple unique routines and HSOPs, resulting in multiple typical daily energy profiles. In comparison to House A, House B's behaviour is very irregular during the winter and summer seasons. Additionally, the behaviours and routines of the occupants are different during the summer and winter seasons, with peaks occurring at different times of the day. This is consistent with the conclusions made for House B and the average summer and winter profiles for the whole study. This household is occupied by two people, with one who works from home who could be relying on the AC to achieve thermal comfort during the day. This would explain the increased number of clusters observed for House B compared to House A. The response to thermal discomfort could be different between the households, resulting in House B having peaks at different times of the day. House A had an individual who worked from home and a student, but the household only exhibited a maximum of four clusters. This household could be stricter with AC usage, not allowing the AC to be

used as much to achieve thermal comfort within the home. This offers an explanation for the differences between Houses A and B.

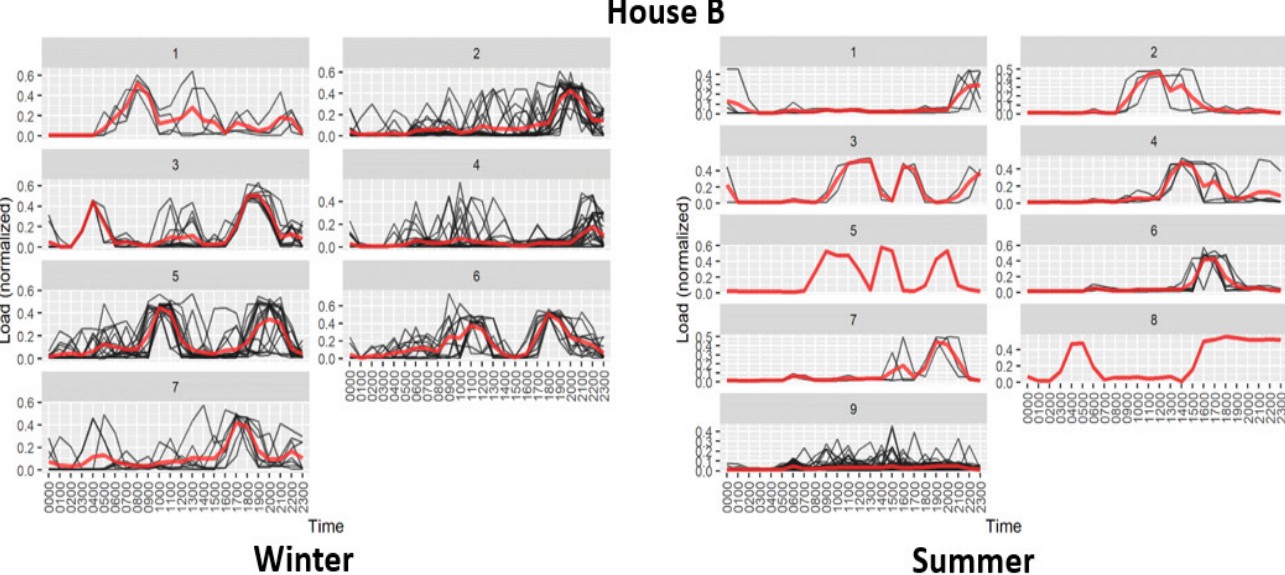

**Figure 9.** House B winter and summer K-mean results. The black lines represent the data objects that were assigned to that cluster and the red lines represent the energy profile for that cluster.

Alternatively, House C follows only two clusters throughout the year, as seen in Figure 6, performing different practices in the summer and winter. The algorithm identified two clusters for the summer and three clusters for the winter period when the dataset was split into these two periods. The occupants consume energy differently in the summer and winter periods, as seen in Figure 10. In the summertime, the occupants consume high amounts of energy during the day, with a sharp peak occurring at 7:00 p.m. and a small peak occurring at 7:00 a.m. During the winter period, there is more variation between the clusters and the shape of each peak. The first cluster features a broad peak occurring throughout the entire day between 5:00 a.m. and 8:00 p.m., which shows the household is occupied during the day for these daily profiles. The second and third clusters are similar, with both showing a dual-peak profile where the peaks occur between 5:00 a.m. and 8:00 a.m. and 4:00 p.m. and 8:00 p.m., respectively. This implies the household is not occupied during the day, which reinforces the variation in the HSOP between these clusters and the first cluster. Alternatively, the adults of the household could restrict the AC usage by the younger individuals to encourage all occupants to perform alternative practices to achieve thermal comfort. This could be a strategy for the household to keep energy bills low with so many occupants. This explains the reduced number of clusters within the home and supports the idea of the relationship between the occupancy of a home and the daily energy profiles.

The comparison between the winter and summer consumption profiles of Houses A, B, and C outlines the variability of the practices performed within each household. The winter and summer practices were identified through clustering analysis and reinforced previous studies' findings that people's lifestyles and behaviours are quite different between these two seasons [83,84]. In the context of the Australian households and this study, people relied on air conditioning more in the winter compared to the summer. Additionally, the variation in their lifestyles during the two seasons was assessed based off the number of clusters identified. House B showed a highly flexible lifestyle during the winter and summer seasons. This household consisted of two occupants, and linking back to the HSOP, the higher variation can be explained by fewer practices being locked into a temporal and spatial field. Lower numbers of clusters were identified in the summer and winter for Houses A and C, which had different family structures compared to House B. Interestingly,

House A showed high variation for their annual cluster analysis discussed earlier in this paper. However, their winter and summer practices follow a routinised lifestyle. This implies the variation observed earlier is a result of other aspects of their lifestyle.

**Figure 10.** House C winter and summer K-mean results. The black lines represent the data objects that were assigned to that cluster and the red lines represent the energy profile for that cluster.

The algorithm demonstrated how each of the three homes consumes energy differently and follows different HSOPs. The initial clustering analysis outlined the different daily routines the households follow throughout the year, with House A and B following several different HSOPs and House C only following three different HSOPs. The number of clusters identified implies variation in the practices and behaviours of the occupants. The household characteristics were used to explain the differences between households by looking at the working lifestyle of each occupant. A higher number of clusters correlated to occupants working from home. However, further investigation into the lifestyles of the occupants is needed to explain these variations in practices. Another finding related to the heating and cooling practices of each home and how occupants rely on the AC to achieve thermal comfort differently. A household can be stricter in their AC usage if they are aiming to reduce their energy bills. Alternatively, they can follow occupancy patterns that do not require heating and cooling often when they are home. This demonstrates the contextual aspect of the way occupants behave and perform practices and how this varies significantly between households.

6. Precinct vs. Individual Household Analysis

This research assessed the energy consumption profiles for the living laboratory as a whole by aggregating the energy data and using the algorithm to identify patterns. This provided an overall sense of how energy is consumed across the 39 homes, where households would synchronise together, resulting in peak demand typically in the morning and afternoon. This synchronisation between households demonstrates how people follow similar routines and perform similar practices at the same time of day. This is due to societal expectations and routines that are followed by all the residents of the precincts that form common energy profiles.

This aggregated precinct analysis provides insight into the overall performance of the homes. However, the minute details of each household are lost when the data are aggregated. When the analysis assessed the energy profiles of individual homes, the algorithm was able to identify more specific patterns and routines that are associated with

that household. For example, House A followed nine different energy profiles, while the precinct analysis observed eight different profiles. House A's energy profiles are more resolved compared to the precinct's profiles. There are more specific peaks in demand that are not identifiable in the precinct's profiles. These specific profiles and timings of peaks for House A are lost in the aggregated analysis.

Additionally, House C only followed two different energy profiles, implying this household consumed energy more consistently compared to the precinct's average (nine different profiles). This demonstrates that House C follows more regular routines and performs everyday practices at the same time throughout the year. The comparison between House A and C has already been discussed; however, this detail relating to the specific household's routines and energy profiles is lost when only considering the precinct as a whole.

This research offers evidence for households to be categorised into groups based off the variation in their energy consumption habits. Referring to Table 2, there were 30 homes that would follow between two and five different energy profiles, while 10 homes would follow between six and nine. This information could be used to separate these homes into two groups: 'low variation' and 'high variation' in consumption. Past research [7] demonstrated classifications and groupings into low and high consumption households. However, this paper continued to classify behaviour and lifestyle impacts on energy consumption to achieve more detailed groupings. This would be beneficial when implementing home energy management systems and automation devices, as they can be designed for homes that follow many different energy profiles or for homes that are more consistent in their consumption. Designing these systems and devices based off data from the precinct as a whole would not consider this variation from house to house, resulting in the potential for these systems to fail by not considering different lifestyles. Taking an average approach to data analysis and not considering grouping homes into clusters can result in information and specific patterns being lost in the analysis.

This finding provides practical insight into future grid management by not assessing precincts as a one system but instead understanding the different household groups that are present within a precinct. These groups can vary based on household characteristics, as well as occupant lifestyles and backgrounds, which impact the way they consume resources. The ability to assess the consistency of energy consumption of each home within a precinct can assist grid managers in grouping homes into categories. One category could be consistent users of energy, such as House C, while another category could contain House A and B, which follow multiple daily routines.

## 5. Conclusions

This paper created a data analytical method to identify and evaluate the different HSOPs a household follows using the household's energy data. Additionally, this analysis was combined with and discussed using psychological theories and social practice theory to explain the shape and nature of the energy profiles generated by the data analysis. This method utilises the k-means machine learning algorithm to classify the energy datasets to identify the different energy profiles a household generally follows throughout the year. This approach successfully identified patterns in the energy data, producing typical daily energy profiles for each household.

The interconnectedness of each occupant's system of practice and how interlinked each practice is affect how flexible the occupants are in their behaviours and routines. The extent of the interconnectedness is measured in this study by the number of different clusters identified by the algorithm. The case studies investigated in this paper show how occupants' lifestyles can result in different daily routines, with House C being more consistent in their consumption routines compared to House A. A general conclusion made from the results of this study is that a higher number of clusters identified results in a higher extent of flexibility by occupants in their routines and energy consumption practices.

Further investigation into the flexibility and lifestyles of the occupants is needed to develop this relationship further.

The results of this paper provide insight into home energy management and tackling the issues of grid energy supply and demand response. The analysis of the heating and cooling practices of Houses A, B, and C demonstrates how each household used their air-conditioning system differently throughout the study period. House B showed much more variation than Houses A and C, indicating that the occupants of this house follow irregular routines and/or require thermal comfort at different times of the day. From this work, the next step is to utilise this approach to predict the daily profiles of households to allow efficient demand response and energy management. Some homes display high variation in their daily profiles, as seen with Houses A and B, which will result in difficulty predicting when they will consume energy on a daily basis. However, homes that are more consistent and follow a couple of different HSOPs can allow energy consumption to be predicted.

The results not only contribute to automation and energy management, but they also directly relate to social practice theory and the development of the HSOP. The algorithm identified multiple daily energy profiles the households follow during the year, which relate to the routines and behaviours of the occupants. Each energy profile represents an HSOP the household follows and provides insight into the individual routines of the occupants.

This paper aimed to group daily energy data into clusters to describe how routinised occupants consume energy within the home. This approach is limited by the accuracy of the k-means algorithm in its ability to categorise the dataset into accurate groups. This accuracy was assessed by visual inspections to confirm the dataset had been split into appropriate clusters. Another limitation of this study is the dataset size, with energy data being collected for almost two years. Larger datasets could provide a more robust analysis, identifying long-term patterns in the households and demonstrating the major routines followed by the occupants. Additionally, the dataset used included the impacts of COVID-19 lockdowns and work-from-home practices, which would have changed the routines and lifestyles of the occupants. Further investigation is required to assess how the virus impacted energy consumption and the behaviours of individuals.

The combination of data analytics and social practice theory can be powerful in explaining how households consume energy throughout the year. This paper demonstrates the effectiveness of bringing these two areas together to provide insight into how occupants consume resources and evaluates the variability in the routines and practices performed by the occupants. The ability to predict this variability and understand how energy profiles will change can be utilised when incorporating renewable energy systems, home automation, and network demand response systems into the residential setting.

**Author Contributions:** Data curation, T.M. Formal analysis, T.M.; Methodology, T.M.; Supervision, J.K.B.; Writing—original draft, T.M.; Writing—review & editing, J.K.B. All authors have read and agreed to the published version of the manuscript.

**Funding:** This research was funded by the Australian Renewable Energy Agency (ARENA) grant number 2018/ARP020 and the APC was funded by Curtin University.

**Informed Consent Statement:** Informed consent was obtained from all subjects involved in the study.

**Data Availability Statement:** The data presented in this study are available on request from the corresponding author. The data are not publicly available due to privacy issues.

**Acknowledgments:** The authors wish to thank the industry partners supporting the project as well as the residents of the homes in this study. Thanks are also given to the wider project team at UTS and Climate-KIC for their collaboration on the Living Laboratory project.

**Conflicts of Interest:** The authors declare no conflict of interest.

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
