# Peer review of "Identifying Home System of Practices for Energy Use with K-Means Clustering Techniques"

_sustainability, doi:10.3390/su14159017_

Round 1
Reviewer 1 Report
1. The title of the research is well-matched and related to the work done. Also, appropriate key studies have been included in this research.
2. In general, the writing of the should be improved.
3. The abstract section is generally good and contains the aims of the work, the summary of the work, and some analysis of the results.
4. From a scientific point of view, this paper presents like those available in the references, except for some minor changes about the household’s energy data, there are many recent references on the topic that have not been taken into consideration, there is not any innovation the formulation and the model are similar to the those available in the literature.
5. There are problems in some references where (Error! Reference source not found) appears in the following lines (279, 328, 369, 371, 386, 412, 486, 527, 535, 548, 571, 635 )
6. The references are sufficient but need to be up-to-date.
7. The control strategy and proposed method are rather simple.
8. The analyses, results, and conclusions in the manuscript were logical for the data used.
9. The topic subject of this paper may be somewhat matched with the topic of the Sustainability journal.
Reviewer 2 Report
1. Abstract can be written in more comprehensive and focused manner. Abstract, summarize the numerical results of proposed work, and discuss how it outperforms existing works. It should not be generalized and specific to the research performed.
2. Contributions should be highlighted in bullet points and justified
3. A ‘Research Gap’ section should incorporate which will states the purpose of the study
4. Comparative analysis with other pattern identification techniques is missing without comparative analysis with the existing literature it is very difficult to find the novelty of the work.
5. Figures can be presented in more better way. Rearrange the x and y axis labels of various figures.
6. Some reference citations are missing such as line number 486.
7. A paragraph related to the practical implementation of the proposed research should be incorporated.
8. Conclusion should be based on the obtained result and performed work.
Reviewer 3 Report
Manuscript ID: Sustainability-1778985
Title: Identifying Home System of Practices for energy use with k-means clustering techniques.
The paper titled “Identifying Home System of Practices for energy use with k-means clustering techniques” employs a data-driven approach to determine the energy demand profile of 39 households. The paper is within the scope of Sustainability. However, there are some weak points in its methodology and analysis. Moreover, the findings seem to be increment and expected. Revision is recommended before considering for publication. Below are some of the specific issues.
Major issues:
1. Review of the k-means clustering technique should be provided for better understanding of the method.
2. Details of the proposed k-means clustering algorithm should be provided.
3. Median and standard deviation of the house design statistics should be added.
4. The normalized load is inconsistent throughout the paper. For instance, the scale of load is 0-0.6 in Figure 3-4, while the others are 0-1.
5. The findings seem increment and expected. More comprehensive analysis should be provided. For example, besides house size and number of occupants, income, ownership of household appliances and electric vehicles have impacts on energy demand.
6. The conclusions include a lot of discussions, making it lengthy. Only important findings and insights should be included.
Minor issues:
7. There are some grammar mistakes. For example, “in response to rise carbon emissions” should be “in response to the rise of carbon emissions” on page 1. Please double-check.
8. There are some problems with the cross-reference of figures.
9. Figure 7’s legend is too small.
Reviewer 4 Report
The paper is well organized and clear. It is recommended to indicate the novelty of your research (how does it differ from other research?) in the Introduction. Also check the links to Figures (Error! Reference source not found).
Author Response
I have included a couple of sentences in the introduction indicating the novelty of this research.
I have fixed up the links to Figures.
Round 2
Reviewer 1 Report
I believe the manuscript has been sufficiently improved to warrant publication in Sustainability.
Author Response
English updated.
Reviewer 2 Report
Authors tried to incorporate most of the comments raised by the reviewer but still the reviewer is not convinced with the comparative analysis. A more detailed and critical comparative analysis is required to show the clear contribution f the proposed work.
Author Response
Added further discussion and comparison in Section 5 : Precinct vs Individual household analysis with more references talking about how this paper challenges other research's conclusions
Reviewer 3 Report
The paper is improved. Suggestions are taken into consideration.
Author Response
No changes required.